# A Spanish Survey on the Perioperative Use of Antimicrobials in Small Animals

**DOI:** 10.3390/ani13152475

**Published:** 2023-07-31

**Authors:** Ignacio Otero Balda, Manuel Fuertes-Recuero, Silvia Penelo Hidalgo, Jorge Espinel Rupérez, Benoit Lapostolle, Tania Ayllón-Santiago, Gustavo Ortiz-Díez

**Affiliations:** 1Department of Small Animal Surgery, Section of Small Animal Clinical Studies, University College Dublin, D04 W6F6 Dublin, Ireland; ignacio.oterobalda@ucd.ie; 2Department of Physiology, College of Veterinary Medicine, Complutense University of Madrid, 28040 Madrid, Spain; manufuer@ucm.es; 3Hospitalization, Emergencies and Critical Care Service, Veterinary Teaching Hospital, Complutense University of Madrid, 28040 Madrid, Spain; spenelo@ucm.es; 4School of Veterinary Medicine, Murdoch University, Perth, WA 6150, Australia; jorge.espinelruperez@murdoch.edu.au; 5Veterinary Teaching Hospital, College of Veterinary Medicine, Alfonso X el Sabio University, 28691 Madrid, Spain; lapostolleben@gmail.com; 6Department of Genetics, Physiology and Microbiology, Faculty of Biological Sciences, Complutense University of Madrid, 28040 Madrid, Spain; 7Department of Animal Medicine and Surgery, College of Veterinary Medicine, Complutense University of Madrid, 28040 Madrid, Spain; gusortiz@ucm.es

**Keywords:** nosocomial infections, questionnaire, veterinarians, small animals, antimicrobial use

## Abstract

**Simple Summary:**

Inappropriate use of antimicrobials during surgeries in small animals can lead to the emergence of drug-resistant bacteria, increased costs and disruption of natural microorganisms. To address this issue, a survey was conducted among Spanish veterinarians to evaluate their current practices regarding antimicrobial use in perioperative settings. The survey revealed that a significant proportion of participants administered antimicrobials before and after surgeries, even in cases where they may not be necessary. Factors such as the level of wound contamination, the patient’s weakened immune system and the use of prostheses influenced antimicrobial selection. Moreover, participants without postgraduate training were more likely to misuse antimicrobials. This highlights the need for evidence-based guidelines and education to ensure proper antimicrobial usage, reducing risks and costs while promoting the overall well-being of animals undergoing surgery.

**Abstract:**

Appropriate use of perioperative antimicrobials can significantly reduce the risk of post-operative infections. However, inappropriate antimicrobial use can result in the creation of multidrug-resistant bacteria, increased costs, host flora disruption, side effects and increased risk of hospital-acquired infections. This survey evaluated the current perioperative use of antimicrobials in small animals by Spanish veterinarians using a web-based questionnaire. Responses were represented using descriptive statistics and a statistical analysis of the association between demographic data and perioperative antimicrobial use was performed. Pre-operative antimicrobials were administered in clean surgery by up to 68.3% of participants, 81.0% in clean-contaminated surgery and 71.3% in dirty surgery, while in the post-operative period, antimicrobials were administered by up to 86.3% of participants in clean surgery, 93.2% in clean-contaminated surgery and 87.5% in dirty surgery. Factors considered “very important” for antimicrobial selection were the degree of wound contamination, patient immunosuppression and use of prosthesis. The most frequently used antimicrobial was beta-lactamase-resistant (or potentiated) penicillin. Post-operative antimicrobial use was associated with participants without specific surgical postgraduate training. This study highlights an overuse of antimicrobials in perioperative procedures in small animal surgery in Spain. Therefore, evidence-based guidelines and further education regarding the correct use of antimicrobial prophylaxis are recommended.

## 1. Introduction

Nosocomial infections, including surgical site infections (SSI), and multi-drug-resistant bacteria constitute an important and growing challenge for human and veterinary medicine. These infections increase morbidity, mortality and hospital stays, especially in Intensive Care Units, which translates into economic losses [1,2,3]. Different veterinary medicine studies specifically evaluating SSI described this complication in 3.0–8.7% of small animal surgeries, with significant variation between different surgical procedures [4,5,6,7,8,9,10]. SSIs lead to several negative outcomes in human and veterinary medicine including tissue destruction, prolonged wound healing, longer hospital stays and increased direct patient costs and mortality [11,12,13].

Appropriate perioperative antimicrobial use can significantly reduce the risk of post-operative SSI and mortality in the human medicine [13,14,15]. However, inappropriate and uncontrolled use of antimicrobial therapies can result in the emergence of multi-drug-resistant bacteria, increased costs, alteration of normal host flora, drug side effects and increased risk of hospital-acquired infections [4,16,17,18,19]. The World Health Organization (WHO) expressed concerns over the current situation surrounding the use of and resistance to antimicrobials, arguing that antimicrobial resistance is a major threat to human health, requiring greater awareness about the adequate prophylactic use of perioperative antimicrobials [20]. Most guidelines for the use of pre-operative antimicrobials in human medicine are based on studies that compare SSI incidence when perioperative antimicrobials are used and the anticipated level of contamination during the surgical procedure (clean, clean-contaminated, contaminated and dirty) [21,22,23,24]. However, guidelines in veterinary medicine [8,13] are usually based on recommendations from current veterinary practice [25,26] and the available human medicine literature [15,22,27].

Some surveys have previously evaluated perioperative antimicrobial use in the veterinary medicine [28,29,30]. A survey performed in the United Kingdom described the use of antimicrobials in clean surgery in up to 25% of small (<1 cm) lumpectomies and 32% of prescrotal castrations [29]. Additionally, 66% of respondents reported administering antimicrobials before surgery [29]. A Colombian survey about antimicrobial use in different clinical contexts reported perioperative antimicrobial use in elective ovariohysterectomies and castrations by up to 86% of participants [28]. One retrospective study in horses undergoing arthroscopic surgery found that perioperative antimicrobial use routinely ignored standard recommendations for perioperative prophylaxis [30].

To the authors’ knowledge, this is the first study evaluating the current use of perioperative antimicrobials in small animal surgery in Spain. Therefore, the aims of this study were: (1) to describe current practices for perioperative antimicrobial use in Spain; (2) to identify factors that influence decision-making about antimicrobial use; (3) to determine the most commonly used antimicrobial agents and their administration route; (4) to compare data for perioperative antimicrobial use with participant demographic data. Additionally, our hypotheses were that (1) overuse of perioperative antimicrobials would be observed with the majority of participants according to the available guidelines [31,32]; (2) decision-making factors considered important by previous surveys would influence participant decision-making regarding antimicrobial use; (3) the most commonly used antimicrobial agent would be the first-generation cephalosporin; (4) less trained participants would be more inclined to administer antimicrobials more often.

## 2. Materials and Methods

A web-based questionnaire (Appendix B and Appendix C) using an online platform (Google Forms^®^) was designed to evaluate the current perioperative use of antimicrobials in small animal surgery. The questionnaire was anonymous and designed to allow only one response per participant, ensuring that each individual could not submit it multiple times. Moreover, it was mandatory for participants to complete the survey using a Google account. The survey was based on a previous questionnaire about perioperative antimicrobial use [29] and the clinical experience of the authors. The questionnaire used for this survey was anonymous, public, freely accessible and had no special incentives, targeting only certain adult members of the public. Initial part of the questionnaire (Appendix B) clarified the purposes of the survey. The survey was designed in a way that ensured the anonymity of participants. We did not request any information that could potentially identify them individually, such as names, addresses, phone numbers or ID numbers [33]. Our primary focus was on collecting aggregated data, compiled with those of other participants, to protect the privacy and confidentiality of each respondent. Moreover, participants provided informed consent by clicking on the designated button to proceed with the survey: “By pressing the “Continue” button, you confirm that you have read the previous information, that you are a small animal clinical veterinarian and that you voluntarily agree to participate in this survey”. Prior to distribution, the questionnaire was tested by 11 small animal surgeons to evaluate its quality and to correct any ambiguous, misleading or inappropriate language.

Although the survey had no time limit, it was designed to be completed in 15 min. Respondents were able to review and change their answers before submitting the questionnaire. The questionnaires were sent three times to each participant from December 2017 to September 2018 through the Association of Spanish Veterinary Specialists in Small Animals (AVEPA) to 5371 registered members, including 2416 males (45.0%) and 2995 females (55.0%).

The questionnaire was divided into three sections: (1) demographic data; (2) perioperative antimicrobial use and factors that influence their use; (3) “agree or disagree” statements related to perioperative antimicrobial use and the emergence of bacterial resistance. 

The first section (Section 1) included questions to ascertain each participant’s demographic information, including gender, university where the respondent obtained their veterinary degree and postgraduate surgical training (non-surgical training, European College of Veterinary Surgeons and American College of Veterinary Surgeons—ECVS/ACVS—diploma, postgraduate masters, postgraduate course, PhD related to small animal surgery), percentage of time dedicated annually to small animal clinical practice (less than and more than 75% of their activity dedicated to small animal veterinary practice), years of experience and percentage of time dedicated annually to small animal surgery (divided into less than and more than 75% of their activity dedicated to surgery). Regarding the center where the respondent practiced professionally, information was gathered on the type of veterinary facility (public or private), geographical region in Spain, total number of veterinarians, number of veterinarians performing surgeries and total number of veterinary assistants.

The second section (Section 2) was further classified In six parts related to the prophylactic use of antimicrobials during the perioperative period and the criteria applied to determine their use. The first part considered the frequency of use (never, rarely, sometimes, usually, always) of pre- and postsurgical antimicrobial therapy for different types of hypothetical surgeries, depending on the degree of contamination. The procedures investigated were the surgeries most commonly performed in small animal practices, classified as clean, clean-contaminated, contaminated and dirty, by the National Research Council (NRC) [21,22,23,24]. The following procedures were included: routine laparotomy ovariohysterectomy in dogs, routine laparotomy ovariohysterectomy in cats, routine orchiectomy in cats, nodulectomy of non-ulcerated 2 cm skin nodules in dogs and closed fracture of the femur with internal fixation in dogs (classified as clean surgery); ovariohysterectomy for open pyometra in dogs, excision of lip mass in dogs and enterotomy for a foreign body and tarsorrhaphy (classified as clean-contaminated); cystotomy with urinary tract infection (considered contaminated surgery) and acute traumatic wound in dogs (classified as dirty surgery). The second part assessed the importance of patient and surgical factors when deciding on antimicrobial use, giving each factor a score from 1 to 5 (1 = not important; 5 = very important). The factors included degree of wound contamination, possibility of evisceration, patient immunosuppression, presence of a drain, use of a prosthesis, acquisition of surgical preparation standards, pre-operative presence of prostheses, impaired physical condition of the patient, surgery time, hollow viscus incision, emergency surgery versus routine surgery, level of clinical experience, hospitalization time and presence of an intravenous catheter. Subsequently, the fourth part of Section 2 contained 12 different classes of antimicrobial agent, including beta-lactamase-resistant (or potentiated) penicillin (e.g., amoxicillin-clavulanic acid), beta-lactamase-sensitive (or non-potentiated) penicillin (e.g., amoxicillin), first-generation cephalosporins (e.g., cefazolin and cephalexin), third-generation cephalosporins (e.g., cefovecin), fluoroquinolones (e.g., enrofloxacin and marbofloxacin), nitroimidazoles (e.g., metronidazol), potentiated sulfonamides (e.g., sulfamethoxazole–trimethoprim), tetracyclines (e.g., doxycycline), macrolides (e.g., erythromycin), lincosamides (e.g., clindamycin), aminoglycosides (e.g., gentamicin and amikacin) and phenicols (e.g., chloramphenicol and florfenicol). Each participant was asked to rank them according to frequency of use from 1 to 12 (1 = least used; 12 = most used). The third part of Section 2 considered the importance of factors determining antimicrobial selection, giving each factor a score from 1 to 5 (1 = not important; 5 = very important). Factors pertaining to antimicrobial choice included potency, activity spectrum, duration of the activity, the intensity of side effects, bactericidal versus bacteriostatic, license for veterinary use, the potential to produce microbial resistance, available routes of administration, cost and shelf life. Additionally, wound location and recommended clinical action protocols were also included. The fifth part considered the administration routes (subcutaneous, intravenous, intramuscular, oral and topical) and time (route not used, before, during and after surgery and post-operative time) for the chosen antimicrobial. Additionally, the sixth section evaluated how frequently a given information source was used, giving a score from 1 to 4 (1 = least used; 4 = most used). These sources included books/drug use guidelines, drug formulary (Vademecum) and drug label (prospectus) and conference proceedings/scientific articles which were consulted to choose the appropriate agent and determine its administration regime.

The third section (Section 3) included 11 agree or disagree statements about issues frequently related to perioperative antimicrobial use, including the effectiveness of pre-operative and post-operative antimicrobial administration in reducing the risk of wound SSI in clean and clean-contaminated surgeries, the effectiveness of pre-operative and post-operative antimicrobial administration reducing the risk of SSI in contaminated surgical wounds, owners’ agreement with the cost of administering antimicrobials, the need for antimicrobial prophylaxis in all surgical procedures and the need for pre-operative and post-operative antimicrobials in all surgical procedures. Additionally, a statement regarding the potential negative impacts of inappropriate antimicrobial use in small animals, leading to bacterial resistance, was included. 

### Statistical Analysis

A statistical analysis was performed to verify any association between the participant demographic data and data for perioperative antimicrobial use. 

Categorical variables were presented as percentages. For continuous variables, data distribution normality was evaluated with the Kolmogorov–Smirnov test. Normal continuous distribution data were presented as a mean (±standard deviation) while non-normal continuous distributions were presented as medians (interquartile range [IQR]). Ordinal data were expressed as percentages, median and IQR. A univariate logistic regression model was performed to compare the demographic data of the participants with the pre- and post-operative antimicrobial use. For statistical analysis, the frequency of use of pre- and post-surgical antimicrobial therapy for different types of hypothetical clean surgeries (including laparotomy ovariohysterectomy in dogs and cats and orchiectomy in dogs and cats) were categorized as never, rarely, sometimes, usually and always; however, due to the low number of cases meeting a particular classification, this variable had to be reclassified as low-frequency (never, rarely and sometimes) and high-frequency (including usually and always). Surgical training was classified as non-surgical training, ECVS/ACVS diploma, postgraduate master, postgraduate course and PhD related to small animal surgery. However, similar to the previous parameter, the low number of participants with postgraduate training meant that this variable had to be reclassified as non-surgical postgraduate training and surgical postgraduate training (ECVS/ACVS diploma, postgraduate master, postgraduate course and PhD related to small animal surgery). Demographic variables including gender (male/female), total number of veterinarians (≤3/>3), number of surgeons out of the total number of veterinarians at the center (≤2/>2) and years of experience (≤14/>14) were obtained. A multivariate regression model was constructed based on the univariate regression model. Variables with a *p*-value < 0.100 in the univariate regression analysis were deemed significant and included in the multivariate logistic regression analysis. The final model was developed using a stepwise forward selection and backward elimination approach. The significance levels for the forward selection and backward elimination steps were set at *p* < 0.050 and *p* < 0.100, respectively. Effect estimates and a 95% confidence interval (CI) were calculated and presented as odds ratio (OR). STATA statistical package (StataCorp, 13.1., College Station, TX, USA) was used for the analysis. A *p*-value of <0.050 was considered statistically significant.

## 3. Results

### 3.1. Demographic Data (Section 1 of the Questionnaire)

Questionnaires were answered by 558 (10.4% response rate) small-animal veterinary practitioners (44.6% males and 55.4% females) throughout Spain. Most participants (99%) had completed their degree at a Spanish university. Fifty-seven percent of the participants had no surgical training, 28.0% had some postgraduate training, 6.5% had a postgraduate Master’s, 5.2% had a PhD related to small animal surgery and 2.7% had an ECVS diploma. Ninety-seven percent of participants worked in centres with more than 75% of their activity dedicated to small animal veterinary practice. The annual percentage of participants dedicated to small animal surgery that dedicate more than 75% of the annual time to surgery was 20.1%. Of the total number of participants, 92.8% of veterinarians worked at a private facility. The highest number of responses was obtained from veterinarians working in Madrid (23.5%) and Catalonia (17.0%). The median number of years of experience of participants was 14 (IQR 7–24 years). The median number of veterinarians working at the respondent’s centre was 3.0 (IQR 2.0–7.5), of whom 2.0 (IQR 2.0–3.0) performed surgery. Additionally, the median number of veterinary technicians working at the respondent’s centre was 2.0 (IQR 1.0–3.0). Demographic data gathered during the study are compiled as Appendix A.

### 3.2. Prophylactic Use of Antimicrobials in the Perioperative Period and Factors That Determine Their Use (Section 2 of the Questionnaire)

#### 3.2.1. Antimicrobial Use in Pre- and Post-Operative Procedures

All participants answered all the questions about antimicrobial use in pre- and post-operative procedures (Table 1). For the hypothetical cases of clean surgery, pre-operative antimicrobials were always used by 44.6–68.3% of participants and never used by 15.1–42.0% of them. Pre-operative antimicrobials were always used by 43.4–81.0% of participants and never used by 2.6–32.4% of respondents in the different clean-contaminated surgeries. Finally, in surgeries considered dirty, pre-operative antimicrobials were always used by 71.3% of participants but never by 8.1%.

By contrast, 34.3–86.3% and 2.4–29.7% of participants reported always or never using post-operative antimicrobials in clean surgeries, respectively. Post-operative antimicrobials were always used by 41.5–93.2% of participants and never used by 1.9–16.6% of respondents in the different clean-contaminated surgeries. In contrast, for contaminated surgeries, 87.5% of participants reported always administering post-operative antibiotics, while only 1.5% never used them. Finally, for dirty surgeries, post-operative antimicrobials were always used by 87.5% of participants and never used by 1.5%.

#### 3.2.2. Relevance of Criteria for Determining Antimicrobial Use

Perioperative factors considered “very important” for antimicrobial selection were the degree of wound contamination, patient immunosuppression and whether the surgical procedure involved using a prosthesis. Factors considered of intermediate importance were the possibility of evisceration, the presence of a drain, surgical preparation standards, impaired physical condition of the patient, surgery, hollow viscus incision and emergency surgery. The factors considered “unimportant” included the presence of an intravenous catheter, length of hospital stay and surgeon’s level of experience (Table 2). 

#### 3.2.3. Antimicrobial Agents and Drug Classes Used

All participants answered all the questions about the different classes of antimicrobial agents used. The most frequently used antimicrobials were beta-lactamase-resistant penicillins, such as amoxicillin-clavulanic acid, followed by first-generation cephalosporins, such as cefazolin or cephalexin. Fluoroquinolones, such as enrofloxacin and marbofloxacin, nitroimidazoles, such as metronidazole, and third-generation cephalosporins, such as cefovecin, constituted the third most used antimicrobials. Beta-lactamase-sensitive penicillins, such as amoxicillin, and tetracyclines, such as doxycycline, were less common. Lincosamides (lincomycin), aminoglycosides (gentamicin and amikacin), phenicols (chloramphenicol and florfenicol), macrolides, such as erythromycin, and potentiated sulphonamides, such as sulfamethoxazole-trimethoprim, represented the least used drugs (Table 3).

#### 3.2.4. Importance of Antimicrobial Characteristics Influencing Antimicrobial Selection

The antimicrobial characteristic that most influenced antimicrobial choice was the spectrum of activity, classified as “very important”, followed by the duration of the activity, the intensity of side effects (median 4 IQR 3–5 n = 554), bactericidal versus bacteriostatic, potential to produce microbial resistance, available administration routes, wound location and recommended clinical action protocols. Less important factors were the license for use in veterinary medicine, cost and half-life (Table 4).

#### 3.2.5. Route and Time of Administration

Of the 558 (100%) participants who answered all the questions, the subcutaneous route for pre-operative antimicrobial administration was selected by 55.9% of respondents, followed by intravenous (31.9%), intramuscular (18.8%), oral (10.9%) and topical (7.3%) routes. Oral administration was the selected post-operative route for 84.1% of participants, followed by topical route (29.2%), with the remaining options presenting negligible rates of application (Appendix A).

#### 3.2.6. Information Source Consulted for Antimicrobial Selection

The evaluation of the information source used for antimicrobial selection and to decide the relevant dosage is presented in Appendix A. 

The main source indicated by respondents to determine dosage recommendations were books and drug use guidelines, followed by drug formulary (Vademecum) and drug label (prospectus), conference proceedings and scientific articles.

### 3.3. Statements Regarding Perioperative Antimicrobial Use (Section 3 of the Questionnaire)

The statements regarding perioperative antimicrobial use and the proportion of respondents who agreed or disagreed with each one are presented in Table 5. 

The survey findings indicated a substantial consensus on the efficacy of antimicrobial use in reducing infection risk in specific surgical scenarios. A considerable majority (ranging from 58.1% to 90.0%) acknowledged the effectiveness of pre-operative and post-operative antimicrobials in decreasing the risk of wound infection in clean surgery, clean-contaminated surgery and contaminated surgical wounds. However, opinions varied concerning general statements about antimicrobial use, such as the administration of prophylaxis when unsure of the need, and the belief in the necessity of the use of pre-operative and post-operative antimicrobials in all surgical procedures. Additionally, nearly all respondents recognized the correlation between inappropriate antimicrobial use in small animals and the development of antibiotic-resistant bacteria.

### 3.4. Demographic Analysis of Perioperative Antimicrobial Use 

Greater use of post-operative antimicrobials was associated with non-surgically postgraduate trained veterinarians, compared to veterinarians with surgical training, for clean surgeries such as canine ovariohysterectomy (adjusted-OR 2.20, CI95% 1.43–3.45, *p* < 0.001), feline ovariohysterectomy (adjusted-OR 2.22, CI95% 1.49–3.33, *p* < 0.001), canine orchiectomy (adjusted-OR 1.89, CI95% 1.30–2.70, *p* = 0.001) and feline orchiectomy (adjusted-OR 1.45, CI95% 1.02–2.08, *p* = 0.040). 

Moreover, statistically significant, but inconclusive associations were found for other demographic variables analyzed and the use of perioperative antimicrobials, such as “Percentage of annual average time dedicated to small animal surgery (%)”, “Surgeons out of the total number of veterinarians in the centre”, “Total number of veterinarians” and “Years of experience” (Appendix A). 

## 4. Discussion

Surgical site infections (SSIs) are a significant concern in veterinary medicine, leading to increased morbidity, mortality and costs [5]. Antibiotic prophylaxis is commonly used to prevent SSIs, but the choice of antibiotics, optimal duration and indication have been debated. A judicious approach, considering patient risk, surgical factors and local antimicrobial susceptibility patterns, is crucial. Although research studies on the use of antibiotics in small animals have been realized [31,34,35,36,37,38,39,40,41,42,43,44,45,46,47,48], there is limited research specifically focused on perioperative antibiotic use in small animals, particularly in Spain [28,29,49].

The main findings of this study were (1) pre-operative antimicrobials were administered in clean surgery by up to 68.3% of participants, 81.0% in clean-contaminated surgery and 71.3% in dirty surgery, while in the post-operative period, antimicrobials were administered by up to 86.3% of participants in clean surgery, 93.2% in clean-contaminated surgery and 87.5% in dirty surgery, (2) factors considered “very important” for antimicrobial selection were the degree of wound contamination, patient immunosuppression and use of a prosthesis; (3) the most frequently used antimicrobials were beta-lactamase-resistant penicillin; (4) post-operative antimicrobial use was associated with participants without specific surgical postgraduate training.

Most participants in this study would administer pre- and post-operative antimicrobials in clean surgery. Classifying surgeries based on contamination level remains controversial [21,22,23,24], particularly in the context of greater surgical complexity [50]. Although limited evidence is available, some studies suggest that the use of pre-operative antimicrobial prophylaxis in clean procedures generates no benefits [4,5,9,10,51,52,53,54,55]. In human medicine, discontinuing antimicrobial administration within 24 h after surgery is recommended [15]. However, in veterinary surgery, there are no evidence-based guidelines informing common practice regarding the duration of antimicrobial use, particularly after orthopedic procedures. Some retrospective studies have reported the potential benefit of post-operative antimicrobial administration [56,57]. However, recent studies suggested no benefit from post-operative antimicrobial administration [5,10,53,58,59]. The present study provides compelling evidence of perioperative antimicrobial overuse when compared to established human CDC guidelines [22,23], the limited perioperative antibiotic guidelines in small animals [31] and the scarce observational studies on risk factors for surgical site infections and antibiotic usage [5,6,7,8,9]. Previous surveys conducted in different countries have also identified a suboptimal use of perioperative antimicrobials in the small animal surgery [28,29,42,46]. Furthermore, the percentage of participants using prophylaxis antimicrobials in our study is higher, especially in feline and canine ovariohysterectomy [29] and similar [28] than previously published surveys. Further clinical research is needed on antimicrobial therapy in small animal surgical procedures, as evidence-based guidelines for perioperative antibiotic use in veterinary medicine are scarce and observational studies are limited.

Factors considered “very important” for perioperative antimicrobial selection by the participants were the degree of wound contamination, patient immunosuppression and use of a prosthesis, as observed by other authors [29]. In addition, the presence of a drain and potential evisceration were considered “very important” factors in other studies [29]. The implantation of a prosthesis is considered a crucial factor, as infections stemming from prostheses can lead to significant complications, including non-resolving infections, the need for subsequent surgical interventions, extended post-operative care, potential financial burdens and possible concerns from pet owners [56,60,61]. The use of post-operative antimicrobials appears to reduce surgical site infections around prosthetics. However, a recent systematic review evaluating post-operative antimicrobial use in dogs following surgery involving the use of a prosthesis (tibial plateau levelling osteotomy) identified insufficient evidence to support its use. Few limitations have been observed in the available literature, including the lack of prospective surveys and the absence of standard treatment protocols [58,59]. Some research studies indicate that the degree of wound contamination and patient immunosuppression are important factors regarding infection and antibiotic use [5,6,7,8,53] contrary to other results [62]. Studies by Espinel et al. (2019), Brown et al. (1997) and Eugster et al. (2004) have established an association between immunosuppression, particularly related to corticosteroid treatment and antibiotic use. The divergent conclusions between studies can be attributed to variations in research design and methodology. Moreover, the cost of the antimicrobial was not deemed a significant factor in decision-making, which could be attributed to the customer’s willingness to accept the necessary price [29,47,63,64]. This contrasts with another study performed in South Africa where cost was identified as one of the main limiting factors for antimicrobial use [44]. Determining important factors for perioperative antimicrobial selection in small animal surgery is challenging due to limited and contradictory literature, warranting further observational studies to explore risk factors and antibiotic usage in surgical site infections. Additionally, effective knowledge transfer through training, seminars, workshops and collaboration with veterinary associations is crucial to ensure responsible antimicrobial practices in the veterinary community.

Considering the widespread use of first-generation cephalosporins as antimicrobial prophylaxis in human medicine [15,22] and companion animals [28,55], and according to the authors’ experience, our initial hypothesis was that this antimicrobial class would be the main choice of veterinarians. However, our study’s findings contradicted this hypothesis, with beta-lactamase-resistant penicillin emerging as the most commonly used antimicrobial prophylaxis, thus rejecting our third hypothesis. This finding is consistent with previous surveys performed in different countries [36,65,66,67,68,69] and it has also been described as antimicrobial prophylaxis in some studies [29,37,38,47,50]. The preference for beta-lactamase-resistant penicillins [70], particularly amoxicillin-clavulanic acid, may be attributed to their historical recommendation for prophylaxis [71]. However, antimicrobial resistance guidelines [72] recommended beta-lactamase-sensitive penicillin (e.g., amoxicillin and ampicillin) or first-generation cephalosporins (e.g., cefalexin) as the preferred choice for prophylaxis over beta-lactamase-resistant penicillin (e.g., amoxicillin-clavulanic acid) to prevent the emergence of greater resistance [73]. Additionally, the incidence of adverse effects, such as hypotension and/or cutaneous signs, appears to be higher with the administration of intravenous amoxicillin-clavulanate than with intravenous cefuroxime for prophylactic antimicrobial therapy in dogs undergoing surgery [74]. First-generation cephalosporins were the second most frequently chosen group of antimicrobials. Other antimicrobials such as third-generation cephalosporins (e.g., cefovecin), fluoroquinolones (e.g., enrofloxacin and marbofloxacin) and nitroimidazoles (e.g., metronidazole) were also chosen by veterinarians in this study as the third most effective antimicrobial prophylaxis, as previous studies [75]. Potentiated penicillins and first-generation cephalosporins are both presently categorized as “Access” by the AWaRE [76] classification from the World Health Organization, but as “Caution” (C) according to EMA’s categorization of antibiotics for use in animals for prudent and responsible use [70] Additionally, it is noteworthy that despite being classified as “Watch” by AWaRE and as “Restrict” (B) in the EMA’s categorization, third-generation cephalosporins and fluoroquinolones are commonly used in this study. We underscore the significance of adhering to the most current guidelines for appropriate antibiotic use in veterinary medicine, which were not available during the inception of our study. These recent developments in clinical guidelines carry crucial implications for promoting judicious antibiotic use, combating antimicrobial resistance on a global scale and safeguarding the health of both animals and humans. Therefore, we emphasize the importance of disseminating and familiarizing the veterinary community with these guidelines to ensure optimal practices and outcomes.

In our study, we observed significant differences in the use of pre- and post-operative antimicrobials by participants with or without postgraduate training in small animal surgery. Due to the differences in the level of training in the group of participating veterinarians with some postgraduate training, these findings must be carefully evaluated, and no reliable conclusions can be drawn from this group. However, participants with no postgraduate training (which constitutes a homogeneous group) used significantly more antimicrobials post-surgery. This difference may be attributed to lower levels of knowledge among participants without postgraduate training, as well as their adherence to the existing recommendations on antimicrobial use. As previously described, veterinary professionals who receive training on antimicrobial control, animal management practices and diagnostic protocols, may be further prepared to make informed decisions about antimicrobial use [35,77]. Consequently, investing in adequate education and training for veterinarians may play a significant role in promoting responsible antimicrobial use in veterinary medicine, making it a critical strategy for mitigating the impact of antimicrobial resistance. Additionally, the implementation of continuing education programs and adherence to clinical guidelines, as supported by some authors [55,78,79], can further promote appropriate antimicrobial use in veterinary medicine. 

The route and timing of antimicrobial administration play important roles in perioperative antimicrobial therapies. In this study, some respondents emphasized the use of the subcutaneous route for pre-operative antimicrobial prophylaxis. This finding is consistent with previous surveys [29]. According to Danish and CDC guidelines [23,31], the ideal initial dose of the most frequently recommended antibiotics (cefazolin) should be administered intravenously 30–60 min before skin incision and repeated at intervals of twice the plasma half-life life [31]. However, the choice of the administration route should consider the bioavailability of the specific product used [80]. Achieving adequate antimicrobial concentrations in both serum and tissue, matching the minimum inhibitory concentration for the most likely microorganisms, is crucial. Depending on the antibiotic employed, the subcutaneous and intramuscular routes may or may not reach peak skin concentration by the start of the surgery. In the latter case, increased antimicrobial-associated morbidity may occur [4]. Unfortunately, the survey did not inquire about which antibiotics were administered by subcutaneous route by the participants, making it challenging to draw definitive conclusions regarding whether adequate concentrations were achieved. As a result, solid conclusions cannot be drawn from this particular finding. 

In our study, the majority of participants most commonly consulted books and guides as their primary information source for decision-making about antimicrobial selection and dosage, which is consistent with other studies in human medicine [81] and small animal veterinary medicine [28]. Nevertheless, this finding differs from another survey conducted on small animals, which identified clinical experience as the primary information source [29]. Moreover, the participants in this study considered the antimicrobial prospectus a useful information source (median score = 3). Antimicrobial prospectus is recommended by agencies such as the European Medicines Agency (EMA) as a reliable source of detailed information [70]. However, veterinarians’ preference for books and guides over prospectus may be attributed to familiarity, convenience or the perception that they provide more up-to-date information.

In our study, the majority of veterinarians acknowledged the link between “the inappropriate use of antimicrobials in small animals” and the development of bacterial resistance [1,2,3], underscoring their awareness of the global threat posed by multi-resistant bacteria. However, despite this recognition, our findings revealed instances of inappropriate antimicrobial use among respondents. To address this issue, we based our evaluation of appropriateness on established guidelines, such as human CDC guidelines [22,23], the limited perioperative antibiotic guidelines in small animals [31] and the available observational studies on risk factors for surgical site infections and antibiotic usage [4,5,6,7,8,9]. Currently, it is imperative to take into account recommendations from both established and newly published guidelines. The observed discrepancies in antimicrobial usage may stem from the lack of comprehensive guidelines and adequate training in antimicrobial stewardship. While veterinarians generally agree on the use of antimicrobials for treating infections, their application in preventive measures remains a subject of debate.

This study represents the first survey conducted to evaluate the current use of antimicrobials in Spain. However, several limitations in our survey methodology should be noted. One of the main limitations was the small number of respondents, which may impact the generalizability of our findings. Additionally, the low response rate introduces a potential selection bias, further affecting the representativeness of the sample. However, this survey targeted a well-selected population of members from AVEPA and the gender distribution of the participants aligned with AVEPA’s registration data at the time of the survey. Regarding the questionnaire designed for the survey, in order to prevent multiple submissions, it was mandatory for participants to have a Google account to complete the survey. Moreover, the current guidelines for appropriate antimicrobial use in animals were not developed and available at the time the questionnaire was conducted, which necessitates interpreting some of the results with caution. Finally, the survey focused on antimicrobials commonly used in veterinary medicine and did not consider other agents used in human medicine, such as carbapenems, which are not recommended for veterinary medicine.

## 5. Conclusions

In this study, a significant majority of veterinarians recognized the link between “the inappropriate use of antimicrobials in small animals” and bacterial resistance. However, our findings also revealed instances of inappropriate antimicrobial use among respondents. We considered established guidelines like human CDC guidelines, limited perioperative antibiotic guidelines in small animals and available observational studies on risk factors for surgical site infections and antibiotic usage to define appropriateness. Additionally, present recommendations from AWARE, EMA and FECAVA for antibiotic selection were taken into account. The observed discrepancies in antimicrobial usage underscore the need for more comprehensive guidelines and adequate training in antimicrobial stewardship. Further research, including observational studies on risk factors for surgical site infections and antimicrobial implications, is essential. Our study emphasizes the importance of addressing inappropriate antimicrobial use through evidence-based guidelines, increased research and enhanced educational efforts for responsible antimicrobial practices in small animal veterinary care.

## Figures and Tables

**Table 1 animals-13-02475-t001:** Frequencies and percentages of veterinarians who use perioperative antimicrobials for ovariohysterectomy and orchiectomy in dogs and cats.

Type of Surgery	Use of Antimicrobials Pre/Post	Frequency (%) of Respondents Who Perform This Surgery					Respondents Who Do Not Perform This Type of Surgery N (%)
		Never	Rarely	Sometimes	Usually	Always	
Clean	Routine laparotomy ovariohysterectomy in dog pre	28.8	9.9	4.5	2.2	54.6	23 (4.1)
	Routine laparotomy ovariohysterectomy in dog post	9.5	9.1	5.2	8.0	68.1	22 (3.9)
Clean	Routine laparotomy ovariohysterectomy in cat pre	33.8	7.9	2.8	2.8	52.6	29 (5.2)
	Routine laparotomy ovariohysterectomy in cat post	12.5	10.0	7.8	8.7	61.1	29 (5.2)
Clean	Routine orchiectomy in dog pre	39.0	6.1	3.5	3.1	48.3	14 (2.5)
	Routine orchiectomy in dog post	16.1	12.3	7.5	8.1	56.0	12 (2.2)
Clean	Routine orchiectomy in cat pre	42.0	4.9	2.9	3.3	46.9	19 (1.8)
	Routine orchiectomy in cat post	29.7	19.7	7.5	8.8	34.3	10 (1.8)
Clean	Excision of a 2-cm, non-ulcerated skin nodule in dog pre	36.2	8.7	4.6	5.9	44.6	16 (2.9)
	Excision of a 2-cm, non-ulcerated skin nodule in dog post	15.4	14.5	15.8	15.3	39.0	14 (2.5)
Clean	Closed fracture of the femur, with internal fixation in dog pre	15.1	4.1	6.3	6.3	68.3	142 825.4)
	Closed fracture of the femur, with internal fixation in dog post	2.4	2.9	2.9	5.5	86.3	143 (25.6)
Clean-contaminated	Ovariohysterectomy for open pyometra in dog pre	2.6	1.8	6.0	8.6	81.0	12 (2.2)
	Ovariohysterectomy for open pyometra in dog post	2.4	0.7	1.5	2.2	93.2	11 (2.0)
Clean-contaminated	Tarsorrhaphy in dog pre	32.4	9.2	8.9	6.0	43.4	112 (20.1)
	Tarsorrhaphy in dog post	16.6	11.4	18.6	11.9	41.5	112 (20.1)
Clean-contaminated	Enterotomy for a foreign body, without discharge of content into the abdominal cavity in dog pre	11.5	6.9	11.1	10.5	60.1	34 (6.1)
	Enterotomy for a foreign body, without discharge of content into the abdominal cavity in dog post	2.9	2.7	3.8	5.9	84.7	34 (6.1)
Clean-contaminated	Excision of lip mass in dog pre	26.8	8.0	9.9	6.1	49.2	32 (5.7)
	Excision of lip mass in dog post	10.1	9.5	15.4	14.5	50.5	33 (5.9)
Contaminated	Cystotomy with urinary tract infection in dog pre	3.7	2.5	5.8	8.3	79.8	39 (7)
	Cystotomy with urinary tract infection in dog post	1.9	0.4	1.4	2.5	93.8	40 (7.2)
Dirty	Surgery for an acute traumatic wound in dog pre	8.1	3.7	8.1	8.8	71.3	14 (2.5)
	Surgery for an acute traumatic wound in dog post	1.5	1.3	5.0	4.8	87.5	15 (2.7)

**Table 2 animals-13-02475-t002:** Number of respondents, percentage and median score of veterinarians who ranked different factors in the decision to use perioperative antimicrobials.

Factors	1 (%)	2(%)	3 (%)	4 (%)	5 (%)	n	Median	25	75
Degree of wound contamination	4 (0.7)	6 (1.1)	18 (3.2)	87 (15.6)	442 (79.4)	557	5.0	5.0	5.0
Possibility of evisceration	69 (12.4)	60 (10.8)	104 (18.7)	139 (25.0)	185 (33.2)	557	4.0	3.0	5.0
Patient immunosuppression	8 (1.4)	13 (2.3)	60 (10.8)	160 (28.7)	316 (56.7)	557	5.0	4.0	5.0
Presence of a drain	14 (2.5)	30 (5.4)	124 (22.3)	193 (34.7)	195 (35.1)	556	4.0	3.0	5.0
Surgery with use of a prosthesis	9 (1.7)	16 (2.9)	67 (12.3)	132 (24.2)	321 (58.9)	545	5.0	4.0	5.0
Surgical preparation standards	28 (5.1)	31 (5.6)	111 (20.1)	125 (22.7)	256 (46.5)	551	4.0	3.0	5.0
Pre-operative presence of prostheses	56 (10.3)	55 (10.1)	149 (27.4)	120 (22.1)	164 (30.1)	544	4.0	3.0	5.0
Impaired physical condition of the patient	22 (3.9)	29 (5.2)	92 (16.5)	177 (31.8)	237 (42.5)	557	4.0	3.0	5.0
Surgery time	23 (4.1)	65 (11.7)	110 (19.8)	143 (25.7)	215 (38.7)	556	4.0	3.0	5.0
Hollow viscus incision	18 (3.3)	36 (6.6)	116 (21.1)	172 (31.3)	207 (37.7)	549	4.0	3.0	5.0
Emergency surgery *versus* routine surgery	45 (8.2)	45 (8.2)	167 (30.4)	145 (26.4)	147 (20.4)	549	4.0	3.0	5.0
Level of clinical experience	91 (16.5)	61 (11.0)	144 (26.0)	144 (26.0)	113 (20.4)	553	3.0	2.0	4.0
Hospitalization time	77 (14.0)	88 (16.0)	178 (32.4)	130 (23.8)	75 (23.8)	548	3.0	2.0	4.0
Presence of an intravenous catheter	129 (23.6)	129 (23.6)	156 (28.6)	80 (14.7)	52 (9.5)	546	3.0	2.0	3.0

**Table 3 animals-13-02475-t003:** Ranking of antimicrobials according to their frequency of use, median and interquartile range. Frequency of antimicrobial use ranged from 1 to 12 (1 = least used; 12 = most used).

Antimicrobials	0 (%)	1 (%)	2 (%)	3 (%)	4 (%)	5 (%)	6 (%)	7 (%)	8 (%)	9 (%)	10 (%)	11 (%)	12 (%)	n	Median	25	75
Beta lactamase resistant penicillins (eg. amoxicillin-clavulanic acid)	7 (1.3)	27 (4.8)	20 (1.8)	11 (2.0)	11 (2.0)	20 (3.6)	25 (4.5)	36 (6.5)	149 (26.7)	5 (0.9)	18 (3.2)	31 (5.6)	208 (37.3)	558	8.0	7.0	12.0
Beta lactamase sensitive penicillins (eg. amoxicillin)	58 (10.4)	135 (24.2)	37 (6.6)	32 (5.9)	30 (5.4)	41 (7.3)	41 (7.3)	37 (6.6)	51 (9.1)	12 (2.2)	22 (3.9)	23 (4.1)	38 (6.8)	558	4.0	1.0	8.0
1st generation cephalosporins (eg Cefazolin, cephalexin)	19 (3.4)	31 (5.6)	24 (4.3)	22 (3.9)	32 (5.7)	37 (6.6)	58 (10.4)	57 (10.2)	88 (15.8)	24 (4.3)	46 (8.2)	66 (11.8)	54 (9.7)	558	7.0	5.0	10.0
3rd generation cephalosporins (e.g., cefovecin)	26 (4.7)	47 (8.4)	51 (9.1)	51 (9.1)	48 (8.6)	53 (9.5)	54 (9.7)	56 (10.0)	41 (7.3)	41 (7.3)	57 (10.2)	22 (3.9)	11 (2.0)	558	6.0	3.0	8.0
Fluoroquinolones (e.g., enrofloxacin, marbofloxacin)	9 (1.6)	28 (5.0)	21 (3.8)	52 (9.3)	53 (9.5)	66 (11.8)	64 (11.5)	45 (8.1)	65 (11.6)	50 (9.0)	48 (8.6)	41 (7.3)	16 (2.9)	558	6.0	4.0	9.0
Nitroimidazoles (e.g., metronidazol)	29 (5.2)	46 (8.2)	27 (4.8)	53 (9.5)	50 (9.0)	62 (11.1)	55 (9.9)	47 (8.4)	78 (14.0)	39 (7.0)	33 (5.9)	24 (4.3)	15 (2.7)	558	6.0	3.0	8.0
Potentiated sulfonamides (e.g., Sulfamethoxazole—trimethoprim)	68 (12.2)	172 (30.8)	78 (14.0)	40 (7.2)	50 (9.0)	41 (7.3)	47 (8.4)	25 (4.5)	14 (2.5)	10 (1.8)	7 (1.3)	4 (0.7)	2 (0.4)	558	2.0	1.0	5.0
Tetracyclines (e.g., doxycycline)	54 (9.7)	120 (21.5)	47 (8.4)	52 (9.3)	50 (9.0)	68 (12.2)	48 (8.6)	43 (7.7)	39 (7.0)	13 (2.3)	12 (2.2)	8 (1.4)	4 (0.7)	558	4.0	1.0	6.0
Macrolides (e.g., erythromycin)	119 (21.3)	262 (47.0)	65 (11.6)	50 (9.0)	18 (3.2)	14 (2.5)	10 (1.8)	12 (2.2)	3 (0.5)	2 (0.4)	1 (0.2)	0 (0)	2 (0.4)	558	1.0	1.0	2.0
Lincosamides (e.g., clindamycin)	82 (14.7)	153 (27.4)	78 (14.0)	56 (10.0)	56 (10.0)	35 (6.3)	35 (6.3)	29 (5.2)	15 (2.7)	9 (1.6)	5(0.9)	5(0.9)	0 (0)	558	2.0	1.0	4.0
Aminoglycosides (e.g., gentamicin, amikacin)	104 (18.6)	216 (38.7)	86 (15.4)	57 (10.2)	24 (4.3)	31 (5.6)	16 (2.9)	7 (1.3)	6 (1.1)	4 (0.7)	4 (0.7)	2 (0.4)	1(0.2)	558	1.0	1.0	3.0
Phenicols (e.g., chloramphenicol, florfenicol)	119 (21.3)	300 (53.8)	58 (10.4)	30 (5.4)	14 (2.5)	10 (1.8)	14 (2.5)	4 (0.7)	5 (0.9)	1 (0.2)	1 (0.2)	0 (0)	2 (0.4)	558	1.0	1.0	1.3

**Table 4 animals-13-02475-t004:** Number of respondents, percentage and median score of veterinarians who ranked various factors in the decision to select a particular antimicrobials perioperatively.

Factors	1 (%)	2 (%)	3 (%)	4 (%)	5 (%)	n	Median	25	75
Antimicrobial potency	16 (2.9)	32 (5.8)	139 (25.1)	205 (37.1)	161 (29.1)	553	4.0	3.0	5.0
Activity spectrum	2 (0.4)	4 (0.7)	20 (3.6)	113 (20.3)	417 (75.0)	556	5.0	4.3	5.0
Duration of activity	19 (3.4)	51 (9.2)	136 (24.6)	174 (31.5)	172 (31.2)	552	4.0	3.0	5.0
Intensity of side effects	12 (2.2)	54 (9.7)	142 (25.6)	171 (30.9)	175 (31.6)	554	4.0	3.0	5.0
Bactericidal *versus* bacteriostatic	42 (7.6)	56 (10.1)	129 (23.4)	190 (34.4)	135 (24.5)	552	4.0	3.0	4.0
The antimicrobial has a license for veterinary use	116 (20.9)	86 (15.5)	119 (21.4)	107 (29.2)	128 (23.0)	556	3.0	2.0	4.0
Potential to produce microbial resistance	46 (8.3)	64 (11.6)	116 (21.0)	135 (24.5)	191 (34.6)	552	4.0	3.0	5.0
Available administration routes	10 (1.8)	27 (4.9)	106 (19.1)	216 (38.9)	196 (35.3)	555	4.0	3.0	5.0
Wound location	38 (6.9)	64 (11.6)	142 (25.6)	175 (31.6)	135 (24.4)	554	4.0	3.0	4.0
Recommended clinical action protocols	23 (4.2)	20 (3.6)	138 (25.0)	217 (39.3)	154 (27.9)	552	4.0	3.0	5.0
Cost	58 (10.5)	92 (16.6)	195 (35.1)	148 (26.7)	62 (11.2)	555	3.0	2.0	4.0
Shelf life	62 (11.2)	89 (16.0)	173 (31.2)	157 (28.3)	74 (13.3)	555	3.0	2.0	4.0

**Table 5 animals-13-02475-t005:** Proportion of respondents who agreed or disagreed with statements regarding perioperative antimicrobial use.

Variable	No (%)	Yes (%)
Pre-operative antimicrobials decrease the risk of wound infection in clean surgery	324 (58.1)	234 (41.9)
Post-operative antimicrobials decrease the risk of wound infection in clean surgery	257 (46.1)	301 (53.9)
Pre-operative antimicrobials decrease the risk of wound infection in clean-contaminated surgery	56 (10.0)	502 (90.0)
Post-operative antimicrobials decrease the risk of wound infection in clean-contaminated surgery	56 (10.0)	502 (90.0)
Pre-operative antimicrobials decrease the risk of infection of a contaminated surgical wound	65 (11.6)	493 (88.4)
Post-operative antimicrobials decrease the risk of infection of a contaminated surgical wound	30 (5.4)	528 (94.6)
The owners agree with the budget that the administration of antimicrobials entails	37 (6.6)	521 (93.4)
I am not sure if antimicrobial prophylaxis is necessary, but I usually prescribe it	269 (48.2)	289 (51.8)
The use of pre-operative antimicrobials is necessary in all surgical procedures	439 (78.7)	119 (21.3)
The use of post-operative antimicrobials is necessary in all surgical procedures	461 (82.6)	97 (17.4)
The inappropriate use of antimicrobials in small animals leads to resistant bacteria	1 (0.2)	557 (99.8)

## Data Availability

The data presented in this study are available in Appendix A and Table 1, Table 2, Table 3, Table 4 and Table 5.

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
