# Peer review of "A Spanish Survey on the Perioperative Use of Antimicrobials in Small Animals"

_animals, 2023, doi:10.3390/ani13152475_

Round 1

Reviewer 1 Report

Many thanks to the authors for the effort and time spent in the preparation of this study. The work is very interesting, both from a clinical and scientific point of view. It meets the required standards and is well written and documented.

1. What is the main question addressed by the research?

The work focused on providing information on the use/misuse of antibiotics by Spanish veterinarians during the perioperative period.

2. Do you consider the topic original or relevant in the field? Does it address a specific gap in the field? Does it address a specific gap in the field? 

This is a widely studied topic in human medicine and of great importance, given the high possibility of creating resistance and the complications that this entails. However, it has not been raised before in the small animal clinic.

3. What does it add to the subject area compared with other published material?

A great reflection on the misuse of antibiotics by many veterinary clinicians. Mainly due to lack of continuing education.

4. What specific improvements should the authors consider regarding the methodology? What futher controls should be considered?

I consider that the paper presents a good methodology and is well structured.

5. Are the conclusions consistent with the evidence and arguments presented and do they address the main question posed?
and address the main issue raised?

Yes

6. Are the references appropriate?
Yes

Author Response

We would kindly like to thank the reviewer for the time and effort dedicated to evaluating this manuscript and his/her positive response.

Reviewer 2 Report

This paper is well-written and highlights once again the gaps (well-known indeed) on antimicrobials when used as surgical prophylaxis, even in small or companion animals; the results reported through questionnaire are always potentially biased by multiple factors; however, these results could be useful to inform also about university lessons, courses or workshops with expert to improve these knowledge gaps.

Minor comments: all the sections are quite long, they could be shortened and become more straightforward. Some results are shown in Tables, while some others are described. Try to avoid too many sentences and keep simple tables when possible.

Overall, an interesting experience that points out, once again, critical misuses.

NA

Author Response

We sincerely express our gratitude to the reviewer for their valuable time and diligent effort in evaluating this manuscript. 

Attached you can find the response to your questions.

Sincerely,

The authors

Reviewer 3 Report

The article is interesting, the structure of the questionnaire is adequate and the sample size is valid.

In my opinion, using Google Forms to administer questionnaires is not a very adequate tool, however I think this choice is due to the simplicity and practicality of use and therefore it is acceptable.

Simple summary

Lines 26-27 Unclear sentence please rephrase (maybe missing noun?)

Key word: please correct the last one

Introduction

Line 72: please check and correct the bibliographic references: works 25 and 26 do not concern human medicine, perhaps they should be placed elsewhere in the sentence

Materials and Methods

Line 106: Could you please specify the meaning of the acronym AVEPA, I didn't find it in the text

Line 133: do you mean only nodules of 2cm or less equal to 2cm?

Line 145 and line 155: Compared to what is reported in appendix 2 the contents of part 3 and part 4 of section 2 are reversed, please check and correct

Results

Line 238 and lines 245-247: If the clean-contaminated category also includes results from the contaminated category can you please add that in the sentence and report the correct result? If the contaminated category is not included, the comment on the results that concern it is missing.

Line 244: The number reported in the text is different from that in the table 2. If the value reported in the table has been rounded, it should be indicated

Line 277: The number reported in the text is different from that in the table 2. If the value reported in the table has been rounded, it should be indicated

Lines 267-279: The text lacks the comment on the use of Lincosamides (e. clindamycin).

Line 287: The number reported in the text is different from that in the table 5. If the value reported in the table has been rounded, it should be indicated.

Line 301: please check and correct the percentage of the "intramuscular" category

Appendix 2

Line 614: Please replace “from 0 to 12” with “from 1 to 12”

Author Response

Thank you for your valuable feedback and insightful comments on our manuscript. We really appreciate your interest in our work, and we sincerely express our gratitude for your valuable time and diligent effort in evaluating this manuscript. 

Attached, you can find the answers to each question. 

We hope that all the concerns have been addressed and that the manuscript has significantly improved after incorporating all the changes suggested.

Sincerely, 

The authors

Reviewer 4 Report

Thank you for submitting this manuscript. The authors present an enormous amount of data about this important topic, which is really complex and difficult to study. I have many comments/suggestions listed below, but the major areas that I think would improve this manuscript would be: 1. to define what you are considering "appropriate" vs. "inappropriate" use in order to support your conclusions (i.e. compare to some sort of guidelines such as WSAVA or the WHO AWaRE categories - or both); 2. to re-evaluate which data are actually important to include in the body of the paper and to discuss (I would consider most of the tables in the body of the paper currently to be good material to put in a supplement, and some of the supplement data better suited to go in the body of the paper), and 3. to contemplate why assess knowledge/attitudes/practices in Spanish veterinarians specifically, or what new information this generates (maybe that is just to find areas to target better education/continuing education - but the relevance of the paper I think could be better defined in the introduction). Overall, this is a great effort. I do wonder if too much data/irrelevant data were included though (not all survey data needs to be reported if it doesn't add to the manuscript in a meaningful way).

- Simple summary: lines 24-25 is not quite a complete sentence, specifically "..and use of influenced antimicrobial selection." This should be rephrased.

- Abstract: line 43: does "beta-lactamase-resistant penicilin" mean a potentiated penicillin? I think the reader will be confused by calling this category of drugs beta-lactamse-resistant drugs.

- Lines 74-76: I would maybe mention whether antimicrobial use in these clean procedures was pre-op, peri-op, post-op etc. since the next line suggests a higher percentage of vets using antimicrobials pre-operatively.

- Lines 88-93: What are these hypotheses based on? For example, when the authors state that "overuse of perioperative antimicrobials would be observed.." is that in comparison to recommendations from WSAVA, something else?

- Line 97: Does the Google Forms system require a Google account? I would state that here so readers would know whether that could confound your results.

- Line 100: Was the survey reviewed by any sort of Institutional Review Board? I am unfamiliar with the approvals processes in Spain, but in many countries there would need to be a review of the survey prior to dissemination for the purposes of publishing the data gained from it. In addition, did the authors have any consultant who specializes in psychometric analysis review the survey?

- Line 106: I would define AVEPA. Same with ECVS/ACVS on line 114 (unless the journal doesn't require it for some reason).

- Line 115 ff: Did you try to target veterinary surgeons specifically, then? For example, if someone responded that they spend <75% of their time dedicated to surgery, were they allowed to fill out the survey? If they were, what was the reasoning behind the 75% cutoff (vs. categories like <25%, 25-50%, >50%-75%, >75%)?

- Line 137: was acute traumatic wound the only "dirty" surgery category? Some would consider that clean-contaminated or contaminated. I wonder if you asked something like "infected wound" or "septic peritonitis" if the % reporting antimicrobials used would be higher, as they are clearly indicated in addition to getting source control. I do think the reported figures in the abstract/results are oddly low for dirty surgery - one would think that nearly 100% of people would use antimicrobials for a surgical procedure involving clearly "dirty" sites, but if the only example given was acute traumatic wound, that could explain a lower % reported.

- Lines 157-160: it would have been interesting to include a small "quiz" of respondents to see if they could correctly identify whether a drug was broad vs. narrow spectrum, duration of action (or just time-dependent vs. concentration-dependent), etc. Can you also explain what is meant by potency?

- Lines 168 ff: was this on a Likert scale, or simply "agree" vs. "disagree?"

- Line 199: I think it's more appropriate to characterize gender as "male/female" - did you have an option for "Other" or "prefer not to say?"

- Line 201: What was the justification for recording years of experience as a dichotomy of </= 14 years vs. >14 years? Did you consider using different categories instead?

- Line 217: the fact that the majority of respondents had no surgical training but said they spend <75% of their time doing surgery is another confounding factor to bring up in the discussion - your results are largely from those with no specific surgical training but do a little surgery. This is perfectly fine, but should be acknowledged as another source of bias in your results (that they may reflect more a general practice population).

- Lines 267-279: It would be interesting to compare whether the drugs used most vs. least frequently correlate with the WHO AWaRE categories (not all of them do - for example, having sulfamethoxazole-trimethoprim as one of the least used drugs is interesting, as it's considered first line or "Access" in the AWaRE category scheme - conversely fluoroquinolones and 3rd gen cephalosporins are not in the Access category).

- Line 290: the "potential to produce microbial resistance" category for example - it would have been nice to confirm that with say the AWaRE categories vs. just the veterinarian's impression (which may or may not be correct).

- Line 313: what is meant by "posology?" Similarly, many readers (myself included) will not know what prospectus and vademecum means - given the international scope of this journal, you may want to explain what those are/mean.

- Lines 326-337: I think reciting in paragraph form the same information relayed in Table 8 is redundant - perhaps just highlight any major discrepancies, surprises, or strong opinions one way or the other.

- Lines 340-350: This is probably the most interesting data in the manuscript - but can you tell us in lines 340-345, postoperative use was higher in non-surgical postgraduate trained vets as compared to all other vets? Or what categories? I think a lot of your response data prior to this section should be in the supplement, and this data in Table S1 should really be in the body of the manuscript (it's your most interesting data).

- Line 358: One thing I've been contemplating reading so far is "is there something that makes Spain different that specifically polling Spanish veterinarians is warranted?" You may want to expand on that a bit either in the intro or discussion.

- Lines 359-363: It's fine to state that, but is that "in accordance" with whatever types of guidelines there are/that are used by many Spanish vets? That's the part that's missing that would give this paper more weight - whether most vets polled actually followed guidelines. This whole paragraph I would separate each (#) out and discuss. For example, for (2), do the authors think these are the most important factors as well, and if not, what are your hypotheses for why most vets think they are?

- Line 367: is the post-op use associated with vets without post-graduate training (i.e. do you think it's related to lack of training?). The way it's phrased in lines 340-341 makes it sound like post-op use was highest in those with some post-graduate training which was non-surgical. Please clarify. If it's associated with lack of training, what do the authors propose would help (more specific classes in vet school? antibiotic specific continuing education requirements? etc.?)

- Line 379: While I don't disagree with you that these data maybe imply overuse of antimicrobials, the rest of the paragraph above implies there are not great data to determine what is proper use - again, why I would suggest comparing to guidelines (WSAVA, AVMA, etc.) - otherwise I don't think you can conclude antibiotics are overused based on your opinion alone (though I personally agree with you).

- Line 391: One could argue that even a surgical site infection could lead to fatal outcomes (septic peritonitis) - do you know for sure that the reason for antibiotic use being important with prostheses was due to concern for fatal outcomes? For example, it is commonplace in the US for surgeons to prescribe post-op antibiotics for something like a TPLO or fracture repair,  but they rarely get infected, and when they do, the patient very very rarely has a fatal outcome (though they do usually need the implant to be removed). So I just struggle with this argument without actually asking the vets if prostheses are important due to fear that the animal will die. I think you might only be able to postulate that it's for fear of death (or maybe more realistically, non-resolving infections requiring a second surgery to be performed, with additional costs, possibly blame from the pet owner, etc.).

- Lines 408-435: This entire paragraph comments on what certain antimicrobials may be used for, rather than commenting on the results of your paper. If you are to include this section in the paper, I think the authors should explain the relevance to the results.

- Lines 436-453: This is a good paragraph! I would try and explain or hypothesize the reason for other results of the paper in a similar fashion to this paragraph.

- Lines 457-460: This is totally dependent on 1. The type and specific antimicrobial and 2. the time between subcutaneous administration and the time of surgery. For example, a pharmacokinetic study in dogs given ceftriaxone IM vs. IV vs. SC showed nearly identical kinetics. So I would argue your statement is true for some drugs and not others. But also the time of administration to surgery needs to be taken into account. Also whether it was truly prophylactic vs. to treat an established infection I would argue matters. So I'm not sure you can conclude much from the route of administration data.

- Line 467: This last sentence contradicts the remainder of the paragraph above, where you just made an argument against subcutaneous administration.

- Line 483: Again, in order to conclude that "inappropriate use" was made by the majority of respondents, you have to define what "appropriate" use is - I think you should go back and decide on your definition of appropriate use for pre- and post-op conditions and reanalyze the data to make it more impactful. Otherwise I do not think you can conclude this without comparing veterinarians' knowledge, attitudes, and practices to some sort of standard.

- Line 500-501: see last comment.

- In addition, there are data in your supplemental documents that I think are better served to be in the body of the paper: for example, Table S1 showed the total # of vets affected frequency of antimicrobial use - that should be in the paper. The # of surgeons out of the total number of vets in the center.

I have no comments for English.

Author Response

We sincerely express our gratitude to the reviewer for their valuable time and diligent effort in evaluating this manuscript. 

Attached you can find the answers to the questions raised. 

We hope that all the concerns have been addressed and that the manuscript has improved after incorporating the changes suggested.

Sincerely,

The authors

Round 2

Reviewer 4 Report

No additional comments - the authors addressed all of my previous comments/questions.